# Towards Manifold Learning of Image-Based Motion Models for Oscillating Vocal Folds

**Sontje Ihler**                                IHLER@IMES.UNI-HANNOVER.DE

**Max-Heinrich Laves**                          LAVES@IMES.UNI-HANNOVER.DE

**Tobias Ortmaier**                             ORTMAIER@IMES.UNI-HANNOVER.DE

*Institute of Mechatronic Systems, Leibniz University Hannover*

**Keywords:** Manifold Learning, Representation Learning, Variational Autoencoder

## 1. Introduction

Our vision is a motion model of the oscillating vocal folds that can prospectively be used for motion prediction and anomaly detection in laryngeal laser surgery during phonation. Surgical procedures on cysts or similar pathologies during phonation promise several medical advantages: the edges of the vocal folds are only visible during phonation, pathologies are squeezed out due to muscle contraction and can presumably be removed with reduced trauma. The main challenge is the high frequency of the phonation ($>100$ Hz).

Shape and motion models are an ongoing field of research in the medical image community (McClelland et al., 2013). A common approach for shape models is based on principal components (PCA) and its variations (Heimann and Meinzer, 2009). PCA is a statistical analysis method commonly used for dimension reduction where high-dimensional data is projected into a low-dimensional representation (embedding). A similar approach was used to compute a motion model for compensation of respiratory motion in radiotherapy using manifold learning based on locally linear embeddings (Baumgartner et al., 2017). An image-based motion model was introduced in 2015 where a PCA of optical flow fields was used to extract high-dimensional, global motion correlations in non-medical image sequences (Wulff and Black, 2015). An autoencoder was used to determine a manifold over the space of human motion (Holden et al., 2015). Equivalent to PCA, an autoencoder computes a lower-dimensional embedding in the bottleneck (latent space). The motion manifold is the subspace of latent representations correlating to expected motion patterns within all possible representations of the latent space. By regularization of the latent space it is possible to compute continuous motion manifolds as the regularization enforces interpolability within the motion subspace.

In this work we propose to learn motion concepts and global motion correlations of the vocal folds and surrounding tissue from endoscopic images using manifold learning based on a variational autoencoder. This has the advantage that training can be performed in a self-supervised manner without applying prior knowledge or additional image or sensor data. The motion model represents a strong prior belief about vocal fold motion.

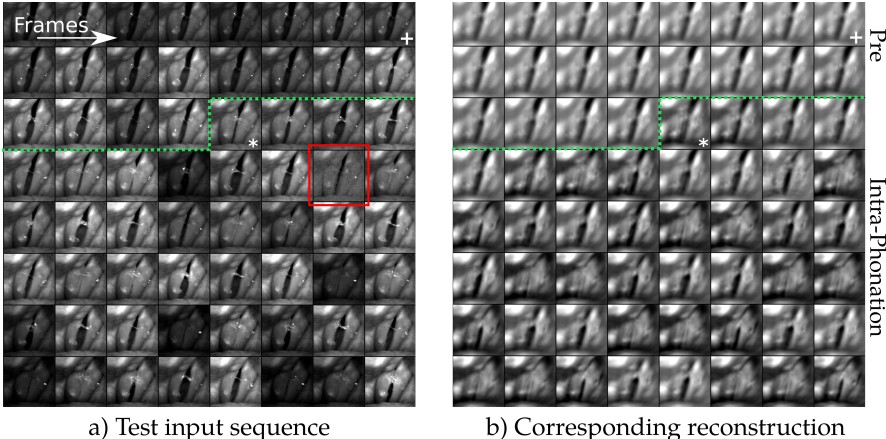

a) Test input sequence         b) Corresponding reconstruction

Figure 1: Excerpt of high-speed sequence of vocal folds during phonation. Green line marks the beginning of visible phonation in image sequence. The red boxes marks corrupted input image. The reconstruction (b) from a VAE shows abstracted concepts of laryngeal movement. [+)] frame 40, [*)] frame 53

## 2. Methods

Autoencoders have shown to be able to learn underlying concepts within data. We enforce learning of abstracted concepts (variations in motion rather than variations in lighting or noise) by choosing a low dimension latent space (bottleneck). To ensure interpolation capability in the motion representation we chose a variational autoencoder (VAE) for probabilistic regularization of the latent space (Kingma and Welling, 2014; Rezende et al., 2014). A VAE is a probabilistic interpretation of an autoencoder based on the principles of Bayes inference. Regularization is achieved by injecting Gaussian noise into the bottleneck. The objective function $L$ is composed of a reconstruction loss $L_{\text{rec}}$ comparing the input $x \in \mathbb{R}^{\text{N}}$ with the reconstructed output $\hat{x} \in \mathbb{R}^{\text{N}}$ and the Kullback-Leibler divergence $L_{\text{kld}}$ for regularizing latent space $z(\mu_{\text{z}}, \sigma_{\text{z}}) \in \mathbb{R}^{\text{M}}$:

$$L = \underbrace{\sum_i^{\text{N}} (x_i - \hat{x}_i)^2}_{L_{\text{rec}}(x,\hat{x})} - \underbrace{\sum_i^{\text{M}} \frac{1}{2} \log(\sigma_i^2) - \mu_i^2 - \sigma_i^2 )}_{L_{\text{kld}}(z)}. \tag{1}$$

Our experiments are based on a high-speed video sequence of vocal folds during phonation (128×128 px @1000 fps). We split the video in disjoint training, validation and test set. Figure 1a shows the first section of the test set. For our VAE implementation we chose latent space dimension M = 32. All layers are fully-connected for high similarity to PCA. The encoder has two hidden layers (each reduces size by 2) and a bottleneck layer with stacked entries for mean $\mu_{\text{z}} \in \mathbb{R}^{32}$ and variance $\sigma_{\text{z}}^2 \in \mathbb{R}^{32}$ required regularization scheme. (For inference $z = \mu_{\text{z}}$.) The decoder is inverse symmetrical to the encoder. We added non-linearity with rectified linear units (ReLU) after each non-bottleneck layer.

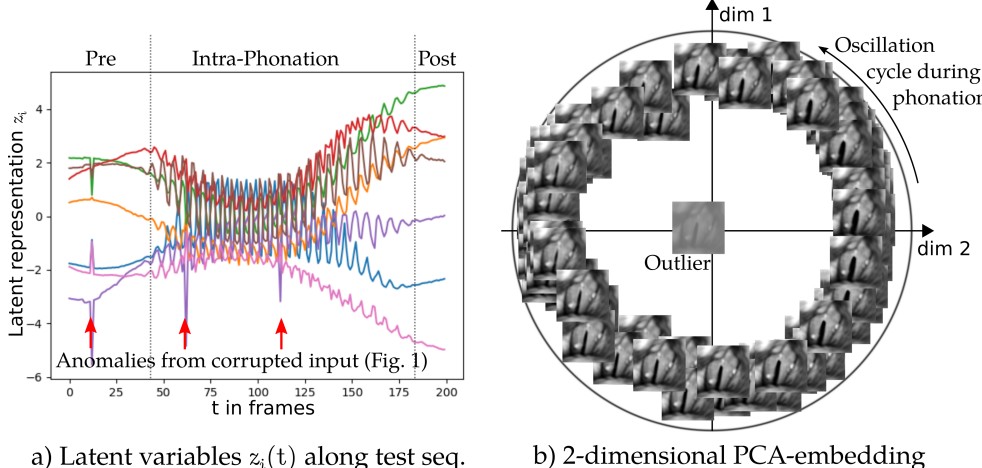

a) Latent variables $z_i(t)$ along test seq.     b) 2-dimensional PCA-embedding

Figure 2: Visualization of latent space representation.

## 3. Results

Figure 1b shows the reconstructions from the learned latent space. It can be seen that the images are reduced to the conceptional basics: the latent representation contains information of how far the vocal folds are apart and it is also possible to distinguish between a relaxed and a contracted larynx (during phonation). The change in brightness was identified as less significant and is not encoded. Figure 2a shows latent variables $z$ over time for the test set in Figure 1a. For better visibility we only show seven components $z_{i \in \{1,...,7\}}$ with highest variance over time. Phonation is made visible by high-frequency oscillations in the latent variables. Hence, the time line can be divided into a pre-phonation, phonation and post-phonation stage. (Note: Visible phonation in the video sequence starts at approx. frame 53, however it becomes visible in the latent space a lot earlier at approx. frame 40.) The high frequency oscillations still seem correlated. Anomalies in the latent space at frame 63 correspond to corrupted input in Figure 1a. The sequence of the latent variables seems structured and presumably suited for prediction tasks. Figure 2b shows an embedding of the latent space. The high-dimensional latent representation is projected into 2d-space using dimensionality reducing PCA. Embedded images correspond to the projected location. Vocal folds on the left are closed and open on the right side. The embeddings are arranged in a circle, making it seem that the embedding uncovers a learned oscillation cycle. Motion forms a continuous circle, i.e. encoded motion seems to be interpolatable.

## 4. Conclusion and Outlook

The proposed method seems to be a promising approach in generating motion and oscillation models of the vocal folds. The current motion model already seems well suited for prediction of movement, as well as for anomaly detection, both essential for patient safety. Deeper assessment of this hypothesis and incorporating more high-level model architectures and regularization techniques are promising steps for further studies.

## Acknowledgments

We thank the Clinic for Phoniatry and Paediatric Audiology at Hannover Medical School for kindly providing us with the laryngoscopic high-speed sequence of vocal folds during phonation. This research has received funding from the European Union as being part of the EFRE OPhonLas project.

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
