# OpenReview forum: "Towards Manifold Learning of Image-Based Motion Models for Oscillating Vocal Folds"
_MIDL.io/2019/Conference/Abstract — MIDL Abstract 2019_

### Official Review · AnonReviewer1 · 2019-04-30
**quantitative results and method details**

**Rating:** 3
**Confidence:** 2

**Review:**

The authors proposed to use Variational Auto-encoders (VAE) to learn motion concepts and motion correlation in a self-supervised manner. Results showed that anomalous frames can be detected by VAE, also motion is continuous in the latent representation which can be used for interpolation. In spite of interesting results, there are the following questions:

1) dimension of the latent space, as described in section 2, is the video sequence of size 128 x 128 x frame number or a single frame of 128 x 128 reduced to a size of 32? The reconstructions seem to be blurry, although it might also be resulted from the limitation of VAE, the reduction from the original size to 32 also loses a lot of information, did the authors ensure that the dimension size is suitable?

2) detection of anomalous frames, as shown in figure 2a, anomalous frames are detected by a value of latent representation, could the authors explain how the y-axis (latent representation) is obtained?

3) quantitative results, as the authors showed two tasks, anomaly detection and possible interpolation between motion frames, is there any quantitative results for them, i.e. precision-recall curve for anomaly detection or euclidean errors for interpolation?

---

### Official Review · AnonReviewer2 · 2019-05-01
**Interesting**

**Rating:** 3
**Confidence:** 3

**Review:**

The work uses a VAE to model to learn representations where PCA and autoencoders have been used. A poster presentation will provide valuable feedback for their project.

The work should compare the VAE representations they learn directly with the representations from the AE and PCA.

They can also try t-SNE on the learned representation as well to see what patterns are visible.

---

### Decision · Program_Chairs · 2019-05-06
**Acceptance Decision**

Accept